# On the Dimensions of Hermitian Subfield Subcodes from Higher-Degree Places

**DOI:** 10.3390/e26050386

**Published:** 2024-04-30

**Authors:** Sabira El Khalfaoui, Gábor P. Nagy

**Affiliations:** 1Institut de Recherche Mathématique de Rennes-IRMAR-UMR 6625, University Rennes, F-35000 Rennes, France; sabiraelkhalfaoui@gmail.com; 2Bolyai Institute, University of Szeged, Aradi Vértanúk tere 1, H-6720 Szeged, Hungary; 3HUN-REN-ELTE Geometric and Algebraic Combinatorics Research Group, Pázmány Péter Sétány 1/C, H-1117 Budapest, Hungary

**Keywords:** Hermitian curves, degree-three places, Riemann–Roch space, Hermitian codes, subfield subcodes, automorphisms of Hermitian codes

## Abstract

The focus of our research is the examination of Hermitian curves over finite fields, specifically concentrating on places of degree three and their role in constructing Hermitian codes. We begin by studying the structure of the Riemann–Roch space associated with these degree-three places, aiming to determine essential characteristics such as the basis. The investigation then turns to Hermitian codes, where we analyze both functional and differential codes of degree-three places, focusing on their parameters and automorphisms. In addition, we explore the study of subfield subcodes and trace codes, determining their structure by giving lower bounds for their dimensions. This presents a complex problem in coding theory. Based on numerical experiments, we formulate a conjecture for the dimension of some subfield subcodes of Hermitian codes. Our comprehensive exploration seeks to deepen the understanding of Hermitian codes and their associated subfield subcodes related to degree-three places, thus contributing to the advancement of algebraic coding theory and code-based cryptography.

## 1. Introduction

The advent of quantum computers presents significant threats to classical cryptographic schemes, requiring the development of post-quantum cryptographic primitives that resist quantum attacks. In this regard, algebraic geometry (AG) codes have gained considerable attention due to their error-correcting capabilities and potential applications in secure communication and cryptographic protocols. Among various classes of AG codes, subfield subcodes stand out against structural attacks, making them good candidates for deployment in post-quantum cryptography.

Within linear codes over finite field extensions, the process of generating subfield subcodes, commonly referred to as restriction, entails converting a given linear code *C* over a large field extension $F_{q^{n}}$ into a code that is defined over a subfield $F_{q^{m}}$, where *m* divides *n*. This strategic approach restricts the codewords of *C* to elements found within the smaller field $F_{q^{m}}$, effectively concealing the details about the structure inherent in *C*. A classic example of this concept is the Reed–Solomon codes, which are algebraic geometry (AG) codes constructed over a projective line. They are widely used in practical applications, with their subfield subcodes represented by Goppa codes. In particular, in cryptography, especially within a McEliece cryptosystem, subfield subcodes play a crucial role in hiding the code structure, thus enhancing its resilience against distinguishing attacks [1,2]. The long-lasting security of the McEliece cryptosystem based on Goppa codes [3] emphasizes its effectiveness in preventing such attacks. Despite subsequent proposals exploring Reed–Solomon codes [4], AG codes, and their subcodes [5], all have been susceptible to structural attacks. By imposing restrictions, cryptographic systems can enhance their security by minimizing the risk of potential attacks aimed at distinguishing the chosen subfield subcode. With growing interest in AG codes, particularly Hermitian codes, they are being evaluated as feasible alternatives to Reed–Solomon codes in specific applications [6]. Hermitian codes have been extensively studied in prior research [7,8,9,10,11,12], particularly those associated with the point at infinity of the Hermitian curve. However, in [13,14], the authors introduced an alternative construction of Hermitian codes associated with higher-degree places on the Hermitian curve.

Our contribution involves conducting further research on Hermitian codes associated with degree-three places, deriving additional properties, and establishing explicit bases for the corresponding Riemann–Roch spaces; additionally, this should align with previous findings in [13]. The stabilizer of a degree-three place has order $3(q^{2}-q+1)$; the action of this group and the associated quotient curve has been studied by Cossidente, Korchmáros, and Torres [15]. We make heavy use of their approach which relates the Hermitian curve with the curve projective curve $XY^{q}+YZ^{q}+ZX^{q}=0$. Beelen, Montanucci, and Vicino [16] studied another class of Hermitian quotient curves, which are obtained by automorphisms stabilizing a degree-three place of the Hermitian curve.

One-point Hermitians of degree-three places have improved minimum distances, as shown by the Matthews–Michel bound [14], and have been further strengthened by Korchmáros and Nagy in [13]. Moreover, we explore the properties of their subfield subcodes, with a particular focus on determining their true dimensions through explicit constructions. This investigation aims to provide a precise understanding of the codes’ capabilities for our future work. Since the family of subfield subcodes of Hermitian codes associated with degree-three places holds promise for the construction of an improved and secure McEliece cryptosystem, the aforementioned investigation will enable a comparison of these parameters with those of other existing codes (see [12], Table 1), such as Goppa codes, to assess the potential improvement in the key size of the McEliece cryptosystem. This suggests that such a proposal could reduce the key size and meet the security level required by NIST [17]. Using bounds on the dimensions offers only an estimate of the code’s performance, which means that this will not help us accurately decide whether these codes can achieve the required security level with an improved key size.

The paper is structured as follows. In Section 2, we introduce the essential background of AG codes constructed from a Hermitian curve, including Hermitian curves, divisors, and the Riemann–Roch space. In Section 3, we provide some facts on the geometry of degree 3 places of the Hermitian curve, and the unitary transformations which stabilize the given degree-three place. Our main tool is the Hermitian sesquilinear form $\langle u,v\rangle =u_{1}v_{1}^{q}-u_{2}v_{3}^{q}-u_{3}v_{2}^{q}$ and the Frobenius map ${\mathrm{Fr}}_{q^{2}}$. Section 4 deals with their corresponding Riemann–Roch spaces. We explore their structure and give explicit and practical bases over $F_{q^{6}}$, and a decomposition into invariant subspaces over $F_{q^{2}}$ (Theorem 3). In Section 5, we study the functional and differential Hermitian codes of a degree 3 place, where we explicitly give the monomial equivalence between them (Theorem 4). In Section 6, we give the main result on the dimensions of the subfield subcodes of degree 3 place Hermitian codes (Theorem 5). This result consists of a theorem that provides a lower bound on the dimensions of the underlying codes, while the conjecture suggests a possible equality based on numerical experiments.

The computational results were obtained using the HERmitian package [18] within the GAP [19] computer algebra system. This involved implementing higher-degree places of Hermitian curves, their divisors and the associated Hermitian codes. This package employs a generic method for computing the bases of Riemann–Roch spaces, independent of the results presented in this paper. Specifically, we acquired computational evidence supporting Conjecture 1 without relying on the theoretical findings of this work.

## 2. Algebraic Geometry (AG) Codes

### 2.1. Hermitian Curves and Their Divisors

For more details, we refer the reader to [15,20,21]. The Hermitian curve, denoted as $H_{q}$, over the finite field $F_{q^{2}}$ in affine coordinates is given by the equation:$$H_{q}:Y^{q}+Y=X^{q+1}.$$
This curve has a genus $g=\frac{q(q-1)}{2}$, classifying it as a maximal curve because it achieves the maximum number of $F_{q^{2}}$-rational points, which is $\#H_{q}\left(F_{q^{2}}\right)=q^{3}+1$. Furthermore, $H_{q}$ has a unique point at infinity, denoted $Q_{\infty }$.

A divisor on $H_{q}$ is a formal sum $D=n_{1}Q_{1}+\cdots +n_{k}Q_{k}$, where $n_{1},\cdots ,n_{k}$ are integers and $Q_{1},\cdots ,Q_{k}$ are points on $H_{q}$. The degree of the divisor *D* is defined as $\deg\left(D\right)=\sum_{i=1}^{k}n_{i}$. The valuation of *D* at a point $Q_{i}$ is $v_{Q_{i}}\left(D\right)=n_{i}$, and the support of *D* is the set $\left\{Q_{i}|n_{i}\neq 0\right\}$.

The Frobenius automorphism, denoted as ${\mathrm{Fr}}_{q^{2}}$, is defined over the algebraic closure ${\bar{F}}_{q^{2}}$ and acts on elements as follows:$${\mathrm{Fr}}_{q^{2}}:{\bar{F}}_{q^{2}}\to {\bar{F}}_{q^{2}},\;x\mapsto x^{q^{2}}.$$

It acts on the points of $H_{q}$ by applying ${\mathrm{Fr}}_{q^{2}}$ to their coordinates. A point *Q* on $H_{q}$ is $F_{q^{2}}$-rational if and only if it is fixed by ${\mathrm{Fr}}_{q^{2}}\left(Q\right)$. Over ${\bar{F}}_{q^{2}}$, the points in $H_{q}$ correspond one-to-one to the places in the function field ${\bar{F}}_{q^{2}}\left(H_{q}\right)$.

For a divisor *D*, its Frobenius image is given by
$${\mathrm{Fr}}_{q^{2}}\left(D\right)=n_{1}{\mathrm{Fr}}_{q^{2}}\left(Q_{1}\right)+\cdots +n_{k}{\mathrm{Fr}}_{q^{2}}\left(Q_{k}\right).$$
and *D* is $F_{q^{2}}$-rational if $D={\mathrm{Fr}}_{q^{2}}\left(D\right)$. In particular, if all points $Q_{1},\ldots ,Q_{k}$ are in $H_{q}\left(F_{q^{2}}\right)$, then *D* is inherently $F_{q^{2}}$-rational.

### 2.2. Riemann–Roch Spaces

For a non-zero function *g* in the function field ${\bar{F}}_{q^{2}}$ and a place *P*, $v_{P}\left(g\right)$ stands for the order of *g* at *P*. If $v_{P}\left(g\right)>0$, then *P* is a zero of *g*, while if $v_{P}\left(g\right)<0$, then *P* is a pole of *g* with multiplicity $-v_{P}\left(g\right)$. The principal divisor of a non-zero function *g* is $\left(g\right)=\sum_{P}v_{P}\left(g\right)P$.

The *Riemann–Roch space* associated with an $F_{q^{2}}$-rational divisor *G* is the $F_{q^{2}}$ vector space
$$L\left(G\right):=\{g\in F_{q^{2}}\left(H_{q}\right)|\left(g\right)+G\geq 0\}\cup 0.$$

From ([20], Riemann’s Theorem 1.4.17), we have
dimL(G)≥deg(G)+1−g,
with equality if deg(G)≥2g−1.

In this work, our primary focus is on an $F_{q^{2}}$-rational divisor *G* of the form sP, where *P* is a degree *r* place in $F_{q^{2}}\left(H_{q}\right)$ and *s* is a positive integer. In the extended constant field $F_{q^{6}}\left(H_{q}\right)$ of $F_{q^{2}}\left(H_{q}\right)$ with degree *r*, let $P_{1},P_{2},\cdots ,P_{r}$ be the extensions of *P*. These points are degree-one places in $F_{q^{2r}}\left(H_{q}\right)$, and, after appropriately labeling the indices, $P_{i}={\mathrm{Fr}}_{q^{2}}^{i}\left(P_{1}\right)$, where the indices are considered modulo *r*.

### 2.3. Hermitian Codes

Here, we outline the construction of an AG code from the Hermitian curve.

In algebraic coding theory, Hermitian codes stand out as a significant class of algebraic geometry (AG) codes, renowned for their distinctive properties. These codes are constructed from Hermitian curves defined over finite fields. These codes are typically viewed as functional AG codes, denoted by $C_{L}\left(D,G\right)$. In this standard approach, the divisor *G* is usually a multiple of a single place of degree one. The set P, which encompasses all the rational points in $H_{q}$, is listed as $\left\{Q_{1},\ldots ,Q_{n}\right\}$. This approach gives rise to a structure known as a one-point code. However, it is important to note that recent research in the field suggests that the use of a more varied selection for the divisor *G* can result in the creation of better AG codes [13,14].

Consider a divisor $D=Q_{1}+Q_{2}+\cdots +Q_{n}$, where all $Q_{i}$ are distinct rational points, and an $F_{q^{2}}$-rational divisor *G* such that Supp(G)∩Supp(D)=∅. By numbering the places in the support of *D*, we define an evaluation map ${\mathrm{ev}}_{D}$ such that ${\mathrm{ev}}_{D}\left(g\right)=\left(g\left(Q_{1}\right),\ldots ,g\left(Q_{n}\right)\right)$ for g∈L(G).

The functional AG code associated with the divisor *G* is
$$C_{L}\left(D,G\right):=\left\{\left(g\left(Q_{1}\right),g\left(Q_{2}\right),\cdots ,g\left(Q_{n}\right)\right)|g\in L\left(G\right)\right\}={\mathrm{ev}}_{D}\left(L\left(G\right)\right),$$

**Theorem 1** ([20], Theorem 2.2.2)**.**
*$C_{L}\left(D,G\right)$ is an [n,k,d] code with parameters*
k=dimL(G)−dimL(G−D)andd≥n−degG.

The dual of an AG code can be described as a residue code (see [20] for more details), i.e.,
$$C_{L}{\left(D,G\right)}^{⊥}=C_{\Omega }\left(D,G\right).$$

Furthermore, the differential code $C_{\Omega }\left(D,G\right)$ is monomially equivalent to the functional code
$$C_{L}\left(D,W+D-G\right),$$
where *W* represents a canonical divisor of ${\bar{F}}_{q^{2}}\left(H_{q}\right)$. The notion of monomial equivalence of codes is defined as follows. Let $C\leq F_{q}^{n}$ be linear subspaces and $\mu =\left(\mu_{1},\ldots ,\mu_{n}\right)\in {\left(F_{q}^{*}\right)}^{n}$ with non-zero entries. We define the Schur product
$$\mu ★C=\{\left(\mu_{1}x_{1},\ldots ,\mu_{n}x_{n}\right)|\left(x_{1},\ldots ,x_{n}\right)\in C\}.$$

The vector μ is also called a multiplier. Clearly, $\mu ★C\leq F_{q}^{n}$. Two linear codes $C_{1},C_{2}\leq F_{q}^{n}$ are monomially equivalent if $C_{2}=\mu ★C_{1}$ for some multiplier μ. Monomially equivalent codes share identical dimensions and minimum distances; however, this correspondence does not preserve all crucial properties of the code.

### 2.4. Subfield Subcodes and Trace Codes

For the efficient construction of codes over $F_{q}$, one approach involves working with codes originally defined over an extension field $F_{q^{m}}$. When considering a code C within $F_{q^{m}}^{n}$, a subfield subcode of C is its restriction to the field $F_{q}$. This process, often employed in the definition of codes such as BCH codes, Goppa codes, and alternant codes, plays a fundamental role.

Let *q* be a prime power and *m* be a positive integer. Let *C* denote a linear code of parameters [n,k] defined over the finite field $F_{q^{m}}$. The *subfield subcode* of *C* over $F_{q}$, represented as ${C|}_{F_{q}}$, is the set
$${C|}_{F_{q}}=C\cap F_{q}^{n},$$
which consists of all codewords in *C* that have their components in $F_{q}$.

The subfield subcode ${C|}_{F_{q}}$ is a linear code over $F_{q}$ with parameters $\left[n,k_{0},d_{0}\right]$, satisfying the inequalities $d\leq d_{0}\leq n$ and $n-k\leq n-k_{0}\leq m\left(n-k\right)$. Moreover, a parity check matrix for *C* over $F_{q}$ provides up to m(n−k) linearly independent parity check equations over $F_{q}$ for the subfield subcode ${C|}_{F_{q}}$. Typically, the minimum distance $d_{0}$ of the subfield subcode exceeds that of the original code *C*.

Let ${\mathrm{Tr}}_{F_{q^{m}}/F_{q}}$ denote the trace function from $F_{q^{m}}$ down to $F_{q}$, expressed as
$${\mathrm{Tr}}_{F_{q^{m}}/F_{q}}\left(x\right)=x+x^{q}+x^{q^{2}}+\ldots +x^{q^{m-1}}.$$
For any vector $c=\left(c_{1},c_{2},\ldots ,c_{n}\right)\in F_{q}^{n}$, we define
$${\mathrm{Tr}}_{F_{q^{m}}/F_{q}}\left(c\right)=\left({\mathrm{Tr}}_{F_{q^{m}}/F_{q}}\left(c_{1}\right),{\mathrm{Tr}}_{F_{q^{m}}/F_{q}}\left(c_{2}\right),\ldots ,{\mathrm{Tr}}_{F_{q^{m}}/F_{q}}\left(c_{n}\right)\right).$$
Furthermore, for a linear code *C* of length *n* and dimension *k* over $F_{q^{m}}$, the code
$${\mathrm{Tr}}_{F_{q^{m}}/F_{q}}\left(C\right)=\left\{{\mathrm{Tr}}_{F_{q^{m}}/F_{q}}\left(c\right)|c\in C\right\}$$
is a linear code of length *n* and dimension $k_{1}$ over $F_{q}$.

A seminal result by Delsarte connects subfield subcodes with trace codes:

**Theorem 2** ([22])**.**
*Let C be an [n,k] linear code over $F_{q}$. Then, the dual of the subfield subcode of C is the trace code of the dual code of C, i.e.,*
$${\left(C{|}_{F_{q}}\right)}^{⊥}={\mathrm{Tr}}_{F_{q^{m}}/F_{q}}\left(C^{⊥}\right).$$

Finding the exact dimension of a subfield subcode of a linear code is typically a hard problem. However, a basic estimation can be obtained by applying Delsarte’s theorem [22]:(1)$${\dim C|}_{F_{q}}\geq n-m\left(n-k\right).$$

In [20] (Chapter 9), various results are discussed with respect to the subfield subcodes and trace codes of AG codes. This motivated us to formulate the following propositions on the dimension of the subfield subcodes of AG codes, which are useful for the case G=sP with a place *P* of higher degree.

**Proposition 1.** 
*Let $G_{1}$ be a positive divisor of the Hermitian curve $H_{q}$ and $D=Q_{1}+\cdots +Q_{n}$ be the sum of $F_{q^{2}}$-rational places such that Supp(G)∩Supp(D)=∅. Assume that $\deg G_{1}<n/q$. Then,*

$$\dim C_{L}\left(D,G_{1}\right){|}_{F_{q}}=1.$$



**Proof.** Let *f* be a function in $L(G_{1})$ such that $f\left(Q_{i}\right)\in F_{q}$ for i=1,⋯,n. Then, $f^{q}-f\in L\left(qG_{1}\right)$ (since $L{\left(G_{1}\right)}^{q}\subseteq L\left(qG_{1}\right)$), and hence $f^{q}-f\in L\left(qG_{1}-D\right)$, where
$$L\left(qG_{1}-D\right)=\ker\left({\mathrm{ev}}_{D}\right)=\left\{x\in L\left(qG_{1}\right)|v_{P_{i}}\left(x\right)>0\;\mathrm{for}\;i=1,\ldots ,n\right\}.$$Since $\deg(qG_{1}-D)<0$, it follows that $L(qG_{1}-D)=0$ and $f^{q}-f=0$, which implies that $f\in F_{q}$. Consequently, $\dim C_{L}\left(D,G_{1}\right){|}_{F_{q}}=1$. □

## 3. The Geometry of Hermitian Degree-Three Places

In this section, we collect useful facts on degree-three places of the Hermitian curve, their stabilizer subgroups, and Riemann–Roch spaces.

### 3.1. The Hermitian Sesquilinear Form

The Hermitian curve $H_{q}$ has the affine equation $X^{q+1}=Y+Y^{q}$. The Hermitian function field ${\bar{F}}_{q^{2}}\left(H_{q}\right)$ is generated by x,y so that $x^{q+1}=y+y^{q}$ holds. The Frobenius field automorphism ${\mathrm{Fr}}_{q^{2}}:x\mapsto x^{q^{2}}$ of the algebraic closure ${\bar{F}}_{q^{2}}$ includes an action on rational functions, places, divisors, and curve automorphisms. For this action, we continue to use the notation ${\mathrm{Fr}}_{q^{2}}$ in the exponent: $P^{{\mathrm{Fr}}_{q^{2}}}$, $f^{{\mathrm{Fr}}_{q^{2}}}$, $D^{{\mathrm{Fr}}_{q^{2}}}$, etc.

Let *K* be a field extension of $F_{q^{2}}$. An affine point is a pair $\left(a,b\right)\in K^{2}$. A projective point (a:b:c) is a one-dimensional subspace {(at,bt,ct)∣t∈K} of $K^{3}$. If c≠0, then the projective point (a:b:c) is identified with the affine point (a/c,b/c). For $u=(u_{1},u_{2},u_{3}),$$v=\left(v_{1},v_{2},v_{3}\right)\in K^{3}$, we define the Hermitian form
$$\begin{array}{c} \langle u,v\rangle =u_{1}v_{1}^{q}-u_{2}v_{3}^{q}-u_{3}v_{2}^{q}. \end{array}$$

Clearly, 〈u,v〉 is additive in *u* and *v*, $\langle \alpha u,\beta v\rangle =\alpha \beta^{q}\langle u,v\rangle$, and
$$\begin{array}{c} {\langle u,v\rangle }^{q}=\langle v^{{\mathrm{Fr}}_{q^{2}}},u\rangle . \end{array}$$

The point *u* is self-conjugate if
$$\begin{array}{c} 0=\langle u,u\rangle =u_{1}^{q+1}-u_{2}u_{3}^{q}-u_{2}^{q}u_{3}. \end{array}$$

This is the projective equation $X^{q+1}-YZ^{q}-Y^{q}Z=0$ of the Hermitian curve $H_{q}$.

Let $u=(u_{1}:u_{2}:u_{3})$ be a projective point. The polar line of *u* has equation
$$u^{⊥}:\langle \left(X_{1},X_{2},X_{3}\right),u\rangle =u_{1}^{q}X_{1}-u_{3}^{q}X_{2}-u_{2}^{q}X_{3}=0.$$
If *u* is on $H_{q}$, then $u^{⊥}$ is the tangent line at *u*. More precisely, $u^{⊥}$ intersects $H_{q}$ at *u* and $u^{{\mathrm{Fr}}_{q^{2}}}$ with multiplicities *q* and 1, respectively. If *u* is $F_{q^{2}}$-rational, then $u=u^{{\mathrm{Fr}}_{q^{2}}}$, and the intersection multiplicity is q+1.

### 3.2. Unitary Transformations and Curve Automorphism

Let *A* be a 3×3 matrix. The linear map u↦uA will also be denoted by *A*. If *A* is invertible, then it induces a projective linear transformation, denoted by $\hat{A}:\left(u_{1}:u_{2}:u_{3}\right)\mapsto \left(u_{1}^{\prime }:u_{2}^{\prime }:u_{3}^{\prime }\right)={\left(u_{1}:u_{2}:u_{3}\right)}^{\hat{A}}$, where
$$\begin{array}{cc} u_{1}^{\prime } & =a_{11}u_{1}+a_{21}u_{2}+a_{31}u_{3},\\ u_{2}^{\prime } & =a_{12}u_{1}+a_{22}u_{2}+a_{32}u_{3},\\ u_{3}^{\prime } & =a_{13}u_{1}+a_{23}u_{2}+a_{33}u_{3}. \end{array}$$

We use the same notation $\hat{A}:\left(X,Y\right)\mapsto \left(X^{\prime },Y^{\prime }\right)={\left(X,Y\right)}^{\hat{A}}$ for the partial affine map:$$\begin{array}{c} \left(X,Y\right)\mapsto \left(X^{\prime },Y^{\prime }\right)=\left(\frac{a_{11}X+a_{21}Y+a_{31}}{a_{13}X+a_{23}Y+a_{33}},\frac{a_{12}X+a_{22}Y+a_{32}}{a_{13}X+a_{23}Y+a_{33}}\right). \end{array}$$

The action $f\left(X,Y\right)\mapsto f({\left(X,Y\right)}^{{\hat{A}}^{-1}})$ of $\hat{A}$ on rational functions will be indicated by $A^{*}$. The following lemma is straightforward.

**Lemma 1.** 
*Let f(X,Y) be a polynomial of total degree n. Define the degree n homogeneous polynomial $F\left(X,Y,Z\right)=Z^{n}f\left(X/Z,Y/Z\right)$. Then,*

$$f^{A^{*}}\left(X,Y\right)=\frac{F(\left(X,Y,1\right)A^{-1})}{{\left(a_{13}X+a_{23}Y+a_{33}\right)}^{n}}.$$



We remark that the line $a_{13}X+a_{23}Y+a_{33}=0$ can be seen as the pre-image of the line at infinity under $\hat{A}$.

The linear transformation *A* is unitary if
〈uA,vA〉=〈u,v〉
holds for all u,v. Since 〈.,.〉 is non-degenerate, unitary transformations are invertible. Moreover, for all u,v, one has
$$\begin{array}{cc} \langle \left(v^{{\mathrm{Fr}}_{q^{2}}}\right)A,uA\rangle & =\langle v^{{\mathrm{Fr}}_{q^{2}}},u\rangle \\ \; & ={\langle u,v\rangle }^{q}\\ \; & ={\langle uA,vA\rangle }^{q}\\ \; & =\langle {\left(vA\right)}^{{\mathrm{Fr}}_{q^{2}}},uA\rangle . \end{array}$$

This implies $\left(v^{{\mathrm{Fr}}_{q^{2}}}\right)A={\left(vA\right)}^{{\mathrm{Fr}}_{q^{2}}}$ for all *v*, that is, *A* and ${\mathrm{Fr}}_{q^{2}}$ commute. This shows that unitary transformations are defined over $F_{q^{2}}$. They form a group which is denoted by GU(3,q). A useful fact is that if $b_{1},b_{2},b_{3}$ is a basis and
$$\langle b_{i}A,b_{j}A\rangle =\langle b_{i},b_{j}\rangle$$
for all i,j∈{1,2,3}, then *A* is unitary.

Let A∈GU(3,q). If (x,y) is a generic point of $H_{q}$, then $\left(x^{\prime },y^{\prime }\right)={\left(x,y\right)}^{\hat{A}}$ satisfies
$${\left(x^{\prime }\right)}^{q+1}-y^{\prime }-{\left(y^{\prime }\right)}^{q}=\langle x^{\prime },y^{\prime }\rangle =\langle x,y\rangle =0.$$
Therefore, $\left(x^{\prime },y^{\prime }\right)$ is a generic point of $H_{q}$, and $A^{*}$ induces an automorphism of the function field ${\bar{F}}_{q^{2}}\left(H_{q}\right)$. If *A* is defined over $F_{q^{2}}$, then $A^{*}$ is an automorphism of $F_{q^{2}}\left(H_{q}\right)$.

### 3.3. Places of Degree Three and Their Lines

Let $a_{1},b_{1}\in F_{q^{6}}\setminus F_{q^{2}}$ be scalars such that $a_{1}^{q+1}=b_{1}+b_{1}^{q}$. In other words, $\left(a_{1},b_{1}\right)$ is an affine point of $H_{q}:X^{q+1}=Y+Y^{q}$, defined over $F_{q^{6}}$. Write $a_{2}=a_{1}^{q^{2}}$, $b_{2}=b_{1}^{q^{2}}$, $a_{3}=a_{2}^{q^{2}}$, $b_{3}=b_{2}^{q^{2}}$, and $p_{i}=\left(a_{i},b_{i},1\right)$. Then, $p_{i+1}=p_{i}^{{\mathrm{Fr}}_{q^{2}}}$, $\langle p_{i},p_{i}\rangle =0$, and
$$\begin{array}{c} 0={\langle p_{i},p_{i}\rangle }^{q}=\langle p_{i}^{{\mathrm{Fr}}_{q^{2}}},p_{i}\rangle =\langle p_{i+1},p_{i}\rangle \end{array}$$
hold for i=1,2,3, with the indices taking modulo three. Since 〈.,.〉 is non-trivial, $\gamma_{i}=\langle p_{i},p_{i+1}\rangle \in F_{q^{6}}\setminus \left\{0\right\}$. More precisely,
$$\begin{array}{c} \gamma_{1}^{q^{3}}={\langle p_{1},p_{2}\rangle }^{q^{3}}={\langle p_{2}^{{\mathrm{Fr}}_{q^{2}}},p_{1}\rangle }^{q^{2}}=\langle p_{2}^{{\left({\mathrm{Fr}}_{q^{2}}\right)}^{2}},p_{1}^{{\mathrm{Fr}}_{q^{2}}}\rangle =\langle p_{1},p_{2}\rangle =\gamma_{1}, \end{array}$$
which shows $\gamma_{i}\in F_{q^{3}}\setminus \left\{0\right\}$. Clearly, $\gamma_{i+1}=\gamma_{i}^{q^{2}}$ and $\gamma_{i+2}=\gamma_{i}^{q}$. By $\gamma_{i}\neq 0$, the vectors $p_{1},p_{2},p_{3}$ are linearly independent over $F_{q^{6}}$.

Let *K* be a field containing $F_{q^{6}}$. Since $p_{1},p_{2},p_{3}$ is a basis in $K^{3}$, any $u\in K^{3}$ can be written as
$$\begin{array}{c} u=x_{1}p_{1}+x_{2}p_{2}+x_{3}p_{3}, \end{array}$$
with $x_{i}\in K$. Computing
$$\begin{array}{c} \langle u,p_{i+1}\rangle =\langle x_{1}p_{1}+x_{2}p_{2}+x_{3}p_{3},p_{i+1}\rangle =x_{i}\langle p_{i},p_{i+1}\rangle , \end{array}$$
we obtain $x_{i}=\langle u,p_{i+1}\rangle /\gamma_{i}$. In the basis $p_{1},p_{2},p_{3}$, the Hermitian form has the shape
$$\begin{array}{cc} \langle u,v\rangle & =\langle x_{1}p_{1}+x_{2}p_{2}+x_{3}p_{3},y_{1}p_{1}+y_{2}p_{2}+y_{3}p_{3}\rangle \\ \; & =x_{1}y_{2}^{q}\langle p_{1},p_{2}\rangle +x_{2}y_{3}^{q}\langle p_{2},p_{3}\rangle +x_{3}y_{1}^{q}\langle p_{3},p_{1}\rangle \\ \; & =\gamma_{1}x_{1}y_{2}^{q}+\gamma_{1}^{q^{2}}x_{2}y_{3}^{q}+\gamma_{1}^{q^{4}}x_{3}y_{1}^{q}. \end{array}$$

In this coordinate frame, the Hermitian curve has projective equation
$$\begin{array}{c} \gamma_{1}X_{1}X_{2}^{q}+\gamma_{1}^{q^{2}}X_{2}X_{3}^{q}+\gamma_{1}^{q^{4}}X_{3}X_{1}^{q}=0. \end{array}$$

Let x,y be the generators of the function field ${\bar{F}}_{q^{2}}\left(H_{q}\right)$ such that $x^{q+1}=y+y^{q}$. Write
$$\begin{array}{c} \ell_{i}=\langle \left(x,y,1\right),p_{i}\rangle =a_{i}^{q}x-y-b_{i}^{q}. \end{array}$$

Then,
$$\begin{array}{c} \left(x,y,1\right)=\frac{\ell_{2}}{\gamma_{1}}p_{1}+\frac{\ell_{3}}{\gamma_{2}}p_{2}+\frac{\ell_{1}}{\gamma_{3}}p_{3} \end{array}$$
and
(2)$$\begin{array}{c} 0=x^{q+1}-y-y^{q}=\langle \left(x,y,1\right),\left(x,y,1\right)\rangle =\frac{\ell_{1}\ell_{2}^{q}}{\gamma_{1}^{q}}+\frac{\ell_{2}\ell_{3}^{q}}{\gamma_{2}^{q}}+\frac{\ell_{3}\ell_{1}^{q}}{\gamma_{3}^{q}}. \end{array}$$

The Hermitian curve $H_{q}$ is non-singular, the places of ${\bar{F}}_{q^{2}}\left(H_{q}\right)$ correspond to the projective points over the algebraic closure ${\bar{F}}_{q^{2}}$. Let $P_{i}$ denote the place corresponding to $\left(a_{i}:b_{i}:1\right)$. $P_{i}$ is defined over $F_{q^{6}}$, $P_{i+1}=P_{i}^{{\mathrm{Fr}}_{q^{2}}}$, and
$$\begin{array}{c} P=P_{1}+P_{2}+P_{3} \end{array}$$
is an $F_{q^{2}}$-rational place of degree three.

The line $a_{i}^{q}X-Y-b_{i}^{q}=0$ is tangent to $H_{q}$ at $p_{i}$; the intersection multiplicities are *q* and 1 at $p_{i}$ and $p_{i+1}$, respectively. This implies that the zero divisor ${\left(\ell_{i}\right)}_{0}$ is $qP_{i}+P_{i+1}$, and the principal divisor of $\ell_{i}$ is
(3)$$\begin{array}{c} \left(\ell_{i}\right)=qP_{i}+P_{i+1}-\left(q+1\right)Q_{\infty }. \end{array}$$

### 3.4. The Stabilizer of a Degree-Three Place

Let $\beta_{1}\in F_{q^{6}}$ be an element such that $\beta_{1}^{q^{3}+1}=1$. Define $\beta_{2}=\beta_{1}^{q^{2}}$, $\beta_{3}=\beta_{2}^{q^{2}}$. Then,
$$\begin{array}{c} \beta_{i}\beta_{i+1}^{q}=\beta_{i}^{q^{3}+1}=1. \end{array}$$

For $p_{i}^{\prime }=\beta_{i}p_{i}$, this implies that
$$\begin{array}{c} \langle p_{i}^{\prime },p_{i+1}^{\prime }\rangle =\beta_{i}\beta_{i+1}^{q}\langle p_{i},p_{i+1}\rangle =\langle p_{i},p_{i+1}\rangle . \end{array}$$

Hence, for all i,j∈{1,2,3},
$$\begin{array}{c} \langle p_{i}^{\prime },p_{j}^{\prime }\rangle =\langle p_{i},p_{j}\rangle . \end{array}$$

This shows that we can extend the map $p_{i}\mapsto p_{i}^{\prime }$ to a unitary linear map $B=B\left(\beta_{1}\right):u\mapsto u^{\prime }$ in the following way. Write
$$\begin{array}{c} u=x_{1}p_{1}+x_{2}p_{2}+x_{3}p_{3}, \end{array}$$
with $x_{i}=\langle u,p_{i+1}\rangle /\gamma_{i}$, and define
(4)$$\begin{array}{c} u^{\prime }=x_{1}p_{1}^{\prime }+x_{2}p_{2}^{\prime }+x_{3}p_{3}^{\prime }=x_{1}\beta_{1}p_{1}+x_{2}\beta_{2}p_{2}+x_{3}\beta_{3}p_{3}. \end{array}$$

The extension *B* is a unique unitary transformation. As we have seen in Section 3.2, this implies that $B=B(\beta_{1})$ is a well-defined element of the general unitary group GU(3,q). The set
$$B=\{B\left(\beta_{1}\right)|\beta_{1}\in F_{q^{6}},\;\beta_{1}^{q^{3}+1}=1\}$$
is a cyclic subgroup of GU(3,q), whose order is $\left|B\right|=q^{3}+1$.

In the projective plane, *B* induces a projective linear transformation $\hat{B}$. $\hat{B}$ is trivial if and only if $\beta_{1}=\beta_{2}=\beta_{1}^{q^{2}}$, that is, if and only if $\beta_{i}\in F_{q^{2}}$. As $\gcd(q^{3}+1,q^{2}-1)=q+1$, $\hat{B}$ is trivial if and only if $\beta_{1}^{q+1}=1$. The set $\hat{B}=\left\{\hat{B}|B\in B\right\}$ is a cyclic group of unitary projective linear transformations, whose order is $|\hat{B}|=q^{2}-q+1$.

In a similar way, we fix the elements
$$\delta_{i}=\gamma_{i}^{\frac{q^{3}-q}{2}}.$$
since $\gamma_{1}\in F_{q^{3}}$, $\delta_{i}\in F_{q^{3}}$. Moreover,
$$\delta_{i}^{q^{3}+1}=\delta_{i}^{2}=\gamma_{i}^{q^{3}-q}=\gamma_{i}^{1-q}.$$

As before, the map
$$\Delta :p_{i}\mapsto p_{i}^{\prime \prime }=\delta_{i}p_{i-1}$$
preserves the Hermitian form:$$\langle p_{i}^{\prime \prime },p_{i+1}^{\prime \prime }\rangle =\langle \delta_{i}p_{i-1},\delta_{i+1}p_{i}\rangle =\delta_{i}^{q^{3}+1}\langle p_{i-1},p_{i}\rangle =\gamma_{i}^{1-q}\gamma_{i-1}=\gamma_{i}.$$

Hence, Δ extends to a unitary linear map, which commutes with ${\mathrm{Fr}}_{q^{2}}$ and normalizes B. Indeed,
$$p_{i}^{\Delta^{-1}B\Delta }={\left(\delta_{i+1}^{-1}p_{i+1}\right)}^{B\Delta }={\left(\delta_{i+1}^{-1}\beta_{i+1}p_{i+1}\right)}^{\Delta }=\beta_{i+1}p_{i},$$
and hence, $\Delta^{-1}B\Delta =B^{q^{2}}$. $\Delta^{3}$ maps $p_{i}$ to $\delta_{1}\delta_{2}\delta_{3}p_{i}$, and
$$\delta_{1}\delta_{2}\delta_{3}=\delta_{1}^{1+q+q^{2}}={\left(\gamma_{1}^{\frac{q^{3}-q}{2}}\right)}^{1+q+q^{2}}={\left(\gamma_{1}^{q^{3}-1}\right)}^{\frac{(q+1)q}{2}}=1.$$

Therefore, Δ has order 3.

As introduced in Section 3.2, the unitary transformations *B* and Δ induce automorphisms $B^{*}$ and $\Delta^{*}$ of the function field.

**Proposition 2.** 
*The group $B^{*}=\left\{B^{*}|B\in B\right\}$ of curve automorphisms has order $q^{2}-q+1$, and $\Delta^{*}$ normalizes $B^{*}$ by*

$${\left(\Delta^{*}\right)}^{-1}B^{*}\Delta^{*}={\left(B^{*}\right)}^{q^{2}}={\left(B^{*}\right)}^{q-1}.$$


*Both $B^{*}$ and $\Delta^{*}$ stabilize the degree-three place P.*


**Proposition 3.** 
*Let $\beta_{1}\in F_{q^{6}}$ be an element such that $\beta_{1}^{q^{3}+1}=1$. Define $\beta_{2}=\beta_{1}^{q^{2}}$, $\beta_{3}=\beta_{2}^{q^{2}}$, and the unitary map $B=B(\beta_{1})\in B$. Then,*

$${\left(\frac{\ell_{i}}{\ell_{i+1}}\right)}^{B^{*}}=\beta_{i}^{q+1}\left(\frac{\ell_{i}}{\ell_{i+1}}\right).$$



**Proof.** By Lemma 1,
$$\begin{array}{cc} \ell_{i}^{B^{*}} & =\frac{\langle \left(x,y,1\right)B^{-1},p_{i}\rangle }{w}\\ \; & =\frac{\langle \left(x,y,1\right),p_{i}B\rangle }{w}\\ \; & =\frac{\langle \left(x,y,1\right),\beta_{i}p_{i}\rangle }{w}\\ \; & =\frac{\beta_{i}^{q}\ell_{i}}{w}, \end{array}$$
where the linear $w=w_{1}x+w_{2}y+w_{3}$ over $F_{q^{2}}$ depends only on *B*. Therefore,
$${\left(\frac{\ell_{i}}{\ell_{i+1}}\right)}^{B^{*}}=\frac{\beta_{i}^{q}}{\beta_{i+1}^{q}}\left(\frac{\ell_{i}}{\ell_{i+1}}\right)=\beta_{i}^{q-q^{3}}\left(\frac{\ell_{i}}{\ell_{i+1}}\right)=\beta_{i}^{q+1}\left(\frac{\ell_{i}}{\ell_{i+1}}\right).$$□

## 4. Riemann–Roch Spaces Associated with a Degree-Three Place

In this section, we keep using the notation of the previous section: $P_{i}$ is a degree-one place of $F_{q^{6}}\left(H_{q}\right)$ associated with the projective point $\left(a_{i}:b_{i}:1\right)$. $P_{i}^{{\mathrm{Fr}}_{q^{2}}}=P_{i+1}$; the index i=1,2,3 always takes modulo three. $P=P_{1}+P_{2}+P_{3}$ is an $F_{q^{2}}$-rational place of degree three of $F_{q^{2}}\left(H_{q}\right)$. The generators x,y of ${\bar{F}}_{q^{2}}\left(H_{q}\right)$ satisfy $x^{q+1}=y+y^{q}$. The rational function $\ell_{i}=a_{i}^{q}x-y-b_{i}^{q}$ is obtained from the tangent line of $H_{q}$ at $P_{i}$.

### 4.1. Basis and Decomposition of the Riemann–Roch Space

Let s,u,v be positive integers such that v≤q and s=u(q+1)−v. Clearly, u,v are uniquely defined by *s*. In [13], the Riemann–Roch space associated with the divisor sP is given as
$$L\left(sP\right)=\left\{\frac{f}{{\left(\ell_{1}\ell_{2}\ell_{3}\right)}^{u}}|f\in F_{q^{2}}\left[X,Y\right],\;\deg f\leq 3u,\;v_{P_{i}}\left(f\right)\geq v\right\}\cup \left\{0\right\}.$$

The Weierstrass semigroup H(P) consists of the integers s≥0 such that the pole divisor ${\left(f\right)}_{\infty }=sP$ for some $f\in F_{q^{2}}\left(H_{q}\right)$, see [20] (Section 6.5) and [16]. If s∉H(P), then it is called a Weierstrass gap; the set of Weierstrass gaps is denoted by G(P). By [13] (Theorem 3.1), we have
G(P)={u(q+1)−v∣0≤v≤q,0<3u≤v}.

By the Weierstrass Gap Theorem ([20], Theorem 1.6.8), |G(P)|=g for a place of degree one. In our case, *P* has degree three and the situation is slightly more complicated.

**Lemma 2.** 

$$3|G\left(P\right)|=\left\{\begin{array}{cc} g & \mathrm{if}\;q\equiv 0,1(\mathrm{mod}\;3),\\ g-1 & \mathrm{if}\;q\equiv 2(\mathrm{mod}\;3). \end{array}\right.$$



**Proof.** The lemma follows from
$$\begin{array}{cc} \left|G(P)\right| & =\sum_{1\leq u\leq q/3}\left|\left\{3u,\ldots ,q\right\}\right|\\ \; & =\sum_{i=1}^{\left\lfloor q/3\right\rfloor }q+1-3u\\ \; & =\frac{\left\lfloor q/3\rfloor (2q-1-3\lfloor q/3\rfloor \right)}{2}. \end{array}$$□

The following proposition gives an explicit basis for the Riemann–Roch space L(sP) over the extension field $F_{q^{6}}$.

**Proposition 4.** 
*Let t,u,v be positive integers such that v≤q and t=u(q+1)−v. Define the rational functions*

$$U_{t,i}=\ell_{i}^{2u-v}\ell_{i+1}^{v-u}\ell_{i+2}^{-u}={\left(\frac{\ell_{i}}{\ell_{i+2}}\right)}^{u}{\left(\frac{\ell_{i+1}}{\ell_{i}}\right)}^{v-u},\;i=1,2,3.$$


*Define $U_{0,i}=1$ as the constant function for i=1,2,3. Then, the following holds:*
*(i)* 
*${\left(U_{t,i}\right)}^{{\mathrm{Fr}}_{q^{2}}}=U_{t,i+1}$.*
*(ii)* 
*The principal divisor of $U_{t,i}$ is*

$$\left(U_{t,i}\right)=-tP+\left((3u-v-1)q+(q-v)\right)P_{i}+\left(v(q-2)+3u\right)P_{i+1}.$$


*In particular, if 3u≥v+1, then $(U_{t,i})\geq -tP$.*
*(iii)* 
*The elements $U_{t,i}$, t≥0, i=1,2,3 are linearly independent with the following exception: q≡2(mod3), $t=(q^{2}-q+1)/3$,*

(5)
$$\begin{array}{c} \frac{U_{t,1}}{\gamma_{1}^{q}}+\frac{U_{t,2}}{\gamma_{2}^{q}}+\frac{U_{t,3}}{\gamma_{3}^{q}}=0. \end{array}$$

*(iv)* 
*The set*

$$U\left(s\right)=\{U_{t,i}|t\in H\left(P\right),\;t\leq s,\;i=1,2,3,\;\left(3t,i\right)\neq \left(q^{2}-q+1,3\right)\}$$


*of rational functions is a basis of L(sP) over $F_{q^{6}}$.*



**Proof.** Note first that u,v are uniquely defined by *t*; therefore, $U_{t,i}$ is well defined. (i) is trivial and (ii) is straightforward from (Equation 3). To show (iii), let us write a linear combination in the form
(6)$$\begin{array}{c} \alpha_{1}U_{t,1}+\alpha_{2}U_{t,2}+\alpha_{3}U_{t,3}=\sum_{\begin{array}{c} r<t\\ i=1,2,3 \end{array}}\lambda_{r,i}U_{r,i} \end{array}$$
such that $\left(\alpha_{1},\alpha_{2},\alpha_{3}\right)\neq \left(0,0,0\right)$. The right-hand side has a valuation of at least −t+1 at $P_{1},P_{2},P_{3}$. If $t\neq (q^{2}-q+1)/3$ and $\alpha_{i}\neq 0$, then the right-hand side has valuation −t at $P_{i+2}$. Hence, $\alpha_{i}=0$ for all i=1,2,3, a contradiction. Assume $t=(q^{2}-q+1)/3$. Then,
$$U_{t,i}=\frac{\ell_{i}\ell_{i+1}^{q}}{{\left(\ell_{1}\ell_{2}\ell_{3}\right)}^{\frac{q+1}{3}}},$$
and (Equation 5) follows from (Equation 2). We can use (Equation 5) to eliminate $U_{t,3}$ from (Equation 6); that is, we can assume $\alpha_{3}=0$. Then, again, the only term that has a valuation −t at $P_{i+2}$ is $\alpha_{i}U_{t,i}$ with $\alpha_{i}\neq 0$. Since the left- and right-hand sides of (Equation 6) must have the same valuations at $P_{1},P_{3}$, $\alpha_{1}=\alpha_{2}=0$ must hold, a contradiction.(iv) By (iii), U(s) consists of linearly independent elements. To show that it is a basis of L(sP), it suffices to show that |U(s)|=dim(L(sP)) for 3s≥2g−2. On the one hand, in this case, dim(L(sP))=3s+1−g. On the other hand,
|U(s)|=1+3(s−|G(P)|)−ε=3s+1−(3|G(P)|+ε),
where ε=0 if q≡0,1(mod3), and ε=1 if q≡2(mod3). By Lemma 2, 3|G(P)|+ε=g, and the claim follows. □

It is useful to have a decomposition of L(sP) over $F_{q^{2}}$.

**Theorem 3.** 
*For a t≥0 integer and $\alpha \in F_{q^{6}}$, define the $F_{q^{2}}$-rational function*

$$W_{t,\alpha }=\alpha U_{t,1}+\alpha^{q^{2}}U_{t,2}+\alpha^{q^{4}}U_{t,3}$$


*and the $F_{q^{2}}$-linear space*

$$W_{t}=\left\{W_{t,\alpha }|\alpha \in F_{q^{6}}\right\}.$$


*For t∈H(P), we have*

$$\dim\left(W_{t}\right)=\left\{\begin{array}{cc} 1 & \mathrm{if}\;t=0,\\ 2 & \mathrm{if}\;q\equiv 2\;(\mathrm{mod}\;3)\;\mathrm{and}\;t=(q^{2}-q+1)/3,\\ 3 & \mathrm{otherwise}. \end{array}\right.$$


*The $F_{q^{2}}$-rational Riemann–Roch space L(sP) has the direct sum decomposition*

(7)
$$\begin{array}{c} L\left(sP\right)=\underset{t\in H(P),\;t\leq s}{⨁}W_{t}. \end{array}$$



**Proof.** For t∈H(P), $W_{t}$ is the set of $F_{q^{2}}$-rational functions in the space spanned by $U_{t,1},U_{t,2},U_{t,3}$. The claims follow from Proposition 4. □

### 4.2. Invariant Subspaces of L(sP)

**Lemma 3.** 
*Let $b\in F_{q^{6}}$ such that $b^{q^{3}+1}=1$. Then, ${\left(b^{q+1}\right)}^{q^{2}}={\left(b^{q+1}\right)}^{q-1}$ and ${\left(b^{q+1}\right)}^{q^{4}}={\left(b^{q+1}\right)}^{-q}$.*


**Proof.** By assumption, $b^{q+1}$ has order $q^{2}-q+1$. The claim follows from the facts that $q^{2}-\left(q-1\right)$ and $q^{4}-q$ are divisible by $q^{2}-q+1$. □

The following lemma shows that the basis elements in U(s) are eigenvectors of $B^{*}$.

**Lemma 4.** 
*Let $\beta_{1}\in F_{q^{6}}$ be an element such that $\beta_{1}^{q^{3}+1}=1$. Define $\beta_{2}=\beta_{1}^{q^{2}}$, $\beta_{3}=\beta_{2}^{q^{2}}$, and the unitary map $B=B(\beta_{1})\in B$. Then,*

$${\left(U_{t,i}\right)}^{B^{*}}=\beta_{i}^{t(q+1)}U_{t,i}.$$



**Proof.** Proposition 3 implies
$${\left(\frac{\ell_{i}}{\ell_{i+2}}\right)}^{B^{*}}=\frac{1}{\beta_{i+2}^{q+1}}\left(\frac{\ell_{i}}{\ell_{i+2}}\right)$$
and
$${\left(\frac{\ell_{i+1}}{\ell_{i}}\right)}^{B^{*}}=\frac{1}{\beta_{i}^{q+1}}\left(\frac{\ell_{i+1}}{\ell_{i}}\right).$$By Lemma 3, $\frac{1}{\beta_{i+2}^{q+1}}={\left(\beta_{i}^{q+1}\right)}^{-q^{4}}={\left(\beta_{i}^{q+1}\right)}^{q}$. Write t=u(q+1)−v with 0≤v≤q. Then,
$$B^{*}:{\left(\frac{\ell_{i}}{\ell_{i+2}}\right)}^{u}{\left(\frac{\ell_{i+1}}{\ell_{i}}\right)}^{v-u}\mapsto {\left(\beta_{i}^{q+1}\right)}^{qu}{\left(\frac{\ell_{i}}{\ell_{i+2}}\right)}^{u}{\left(\beta_{i}^{q+1}\right)}^{-v+u}{\left(\frac{\ell_{i+1}}{\ell_{i}}\right)}^{v-u}$$The result follows from the definition of *u* and *v*. □

**Proposition 5.** 
*(i)* 
*Let $\beta_{1}\in F_{q^{6}}$ be an element such that $\beta_{1}^{q^{3}+1}=1$, and $B=B(\beta_{1})\in B$. Then,*

$${\left(W_{t,\alpha }\right)}^{B^{*}}=W_{t,\beta_{1}^{t(q+1)}\alpha }.$$

*(ii)* 
*The subspaces $W_{t}$, t∈H(P) are $B^{*}$-invariant.*
*(iii)* 
*The $F_{q^{2}}B^{*}$-modules $W_{t}$ and $W_{s}$ are isomorphic if and only if one of the following holds:*
*(a)* 
*$s\equiv t\;(\mathrm{mod}\;q^{2}-q+1)$;*
*(b)* 
*$s\equiv \left(q-1\right)t\;\left(\mathrm{mod}\;q^{2}-q+1\right)$;*
*(c)* 
*$s\equiv -qt\;(\mathrm{mod}\;q^{2}-q+1)$.*



**Proof.** (i) and (ii) follow from Lemma 4. (iii) Let $\Phi :W_{t}\to W_{s}$ be an $F_{q^{2}}B^{*}$-module isomorphism between $W_{t}$ and $W_{s}$. It can be written as
$${\left(W_{t,\alpha }\right)}^{\Phi }=W_{t,\alpha \varphi },$$
where $\varphi :F_{q^{6}}\to F_{q^{6}}$ is an $F_{q^{2}}$-linear bijection. Moreover,
$$\begin{array}{cc} {\left(W_{t,\alpha }\right)}^{B^{*}\Phi } & ={\left(W_{t,\beta_{1}^{t(q+1)}\alpha }\right)}^{\Phi }=W_{s,(\beta_{1}^{t(q+1)}\alpha )\varphi },\\ {\left(W_{t,\alpha }\right)}^{\Phi B^{*}} & ={\left(W_{s,\alpha \varphi }\right)}^{B^{*}}=W_{s,\beta_{1}^{s(q+1)}\left(\alpha \varphi \right)}. \end{array}$$Since $b=\beta_{1}^{q+1}$ satisfies $b^{q^{2}-q+1}=1$, this means that for any $\alpha ,b\in F_{q^{6}}$, $b^{q^{2}-q+1}=1$, we have
$$\left(b^{t}\alpha \right)\varphi =b^{s}\left(\alpha \varphi \right).$$Let *b* be an element of order $q^{2}-q+1$ in $F_{q^{6}}$. If $b^{t}$ or $b^{s}$ is in $F_{q^{2}}$, then $b^{t}=b^{s}$ and a) hold. Assume that neither $b^{t}$ nor $b^{s}$ is in $F_{q^{2}}$. Then, $F_{q^{6}}=F_{q^{2}}\left(b^{t}\right)=F_{q^{2}}\left(b^{s}\right)$, and over $F_{q^{2}}$, the minimal polynomial of $b^{t}$ has the degree three. Assume $b^{3t}+c_{1}b^{2t}+c_{2}b^{t}+c_{3}=0$ with $c_{0},c_{1},c_{2}\in F_{q^{2}}$. Then,
$$\begin{array}{cc} 0 & =(b^{3t}+c_{1}b^{2t}+c_{2}b^{t}+c_{3})\varphi \\ \; & =\left(b^{3t}\varphi \right)+c_{1}\left(b^{2t}\varphi \right)+c_{2}\left(b^{t}\varphi \right)+c_{3}\left(1\varphi \right)\\ \; & =\left(b^{3s}+c_{1}b^{2s}+c_{2}b^{s}+c_{3}\right)\left(1\varphi \right). \end{array}$$As φ is bijective, 1φ≠0, $0=b^{3s}+c_{1}b^{2s}+c_{2}b^{s}+c_{3}$ follows. This means that $b^{s}$ has the same minimal polynomial and $b^{t}\to b^{s}$ extends to a field automorphism of $F_{q^{6}}$ over $F_{q^{2}}$. This implies $b^{s}=b^{t}$, $b^{s}={\left(b^{t}\right)}^{q^{2}}$ or $b^{s}={\left(b^{t}\right)}^{q^{4}}$, and the claim follows. □

## 5. Hermitian Codes of Degree-Three Places and Their Duals

In this section, we explore the one-point Hermitian codes of degree-three places and their dual codes. Let *P* be a degree-three place on the Hermitian curve $H_{q}$; $Q_{1},\ldots ,Q_{n},Q_{\infty }$ are its $F_{q^{2}}$-rational places, where $n=q^{3}$. We define the divisors $D=Q_{1}+Q_{2}+\cdots +Q_{n}$, $\tilde{D}=D+Q_{\infty }$, and G=sP for a positive integer *s*.

### 5.1. Functional Hermitian Codes of Degree-Three Places

Given a divisor *D* and *G*, we define the degree-three place functional Hermitian code $C_{L}\left(D,sP\right)$ as:$$C_{L}\left(D,G\right):=\left\{\left(g\left(Q_{1}\right),g\left(Q_{2}\right),\cdots ,g\left(Q_{n}\right)\right)\;\;|\;\;g\in L\left(G\right)\right\},$$

This code forms an [n,k] AG code, where k≥3s−g+1, achieving equality when $\lfloor \frac{2g-2}{3}\rfloor <s<n/3$. Furthermore, the code has a minimum distance $d\geq d^{*}=q^{3}-3s$, where $d^{*}$ is the designed minimum distance.

Furthermore, another degree-three place functional Hermitian code associated with *G*, denoted by $C_{L}\left(\tilde{D},G\right)$, is constructed by evaluating the functions in L(G) at all rational points $Q_{1},Q_{2},\cdots ,Q_{n}$ and the point at infinity $Q_{\infty }$ as follows:$$C_{L}\left(\tilde{D},G\right):=\left\{\left(g\left(Q_{1}\right),g\left(Q_{2}\right),\cdots ,g\left(Q_{n}\right),g\left(Q_{\infty }\right)\right)\;\;|\;\;g\in L\left(G\right)\right\},$$

Clearly, $C_{L}\left(\tilde{D},G\right)$ has a length of n+1. Concerning the dimensions, we have the following result.

**Proposition 6.** 
*If $s<q^{3}/3$, then L(sP), $C_{L}\left(D,G\right)$ and $C_{L}\left(\tilde{D},G\right)$ have the same dimensions.*


**Proof.** If $f\in \ker{\mathrm{ev}}_{D}$, then f∈L(sP−D), which is trivial if $s<q^{3}/3$. In this case, $\ker{\mathrm{ev}}_{\tilde{D}}$ is also trivial. □

**Remark 1.** 
*Numerical experiments show that L(sP), $C_{L}\left(D,G\right)$ and $C_{L}\left(\tilde{D},G\right)$ have the same dimension if $s<(q^{3}+1)/3+q-1$.*


In the study of the divisors *D* and $\tilde{D}$, we make use of the polynomial
$$R\left(X,Y\right)=X\prod_{\begin{array}{c} c\in F_{q^{2}}\\ c^{q}+c\neq 0 \end{array}}\left(Y-c\right).$$
As shown in [13] (Section 2), the principal divisor of $R\left(x,y\right)\in F_{q^{2}}\left(H_{q}\right)$ is
(8)$$\begin{array}{c} \left(R\left(x,y\right)\right)=D-q^{3}Q_{\infty }. \end{array}$$

Further properties of R(x,y) are given in the following proposition.

**Proposition 7.** 
*In the function field, we have*

$$x^{q}R\left(x,y\right)=y^{q^{2}}-y\;\mathrm{and}\;R\left(x,y\right)=x^{q^{2}}-x.$$


*The differential of R(x,y) is*

d(R(x,y))=−dx.



**Proof.** Clearly,
$$\prod_{\begin{array}{c} c\in F_{q^{2}}\\ c^{q}+c=0 \end{array}}\left(Y-c\right)=Y^{q}+Y,$$
and
$$\prod_{\begin{array}{c} c\in F_{q^{2}}\\ c^{q}+c\neq 0 \end{array}}\left(Y-c\right)=\frac{\prod_{c\in F_{q^{2}}}\left(Y-c\right)}{\prod_{\begin{array}{c} c\in F_{q^{2}}\\ c^{q}+c=0 \end{array}}\left(Y-c\right)}=\frac{Y^{q^{2}}-Y}{Y^{q}+Y}.$$Hence, by $x^{q+1}=y+y^{q}$,
$$x^{q}R\left(x,y\right)=x^{q+1}\prod_{\begin{array}{c} c\in F_{q^{2}}\\ c^{q}+c\neq 0 \end{array}}\left(y-c\right)=x^{q+1}\frac{y^{q^{2}}-y}{y^{q}+y}=y^{q^{2}}-y.$$Using this, we obtain
$$\begin{array}{c} x^{q}\left(x^{q^{2}}-x\right)={\left(x^{q+1}\right)}^{q}-x^{q+1}=y^{q}+y^{q^{2}}-\left(y+y^{q}\right)=y^{q^{2}}-y=x^{q}R\left(x,y\right). \end{array}$$
Canceling by $x^{q}$, we get $R\left(x,y\right)=x^{q^{2}}-x$, and d(R(x,y))=−dx follows immediately. □

### 5.2. Differential Hermitian Codes of Degree-Three Places

Differential Hermitian codes of degree-three places are essential counterparts to functional codes on the Hermitian curve $H_{q}$. The dual code $C_{\Omega }\left(D,G\right)$ of $C_{L}\left(D,G\right)$ is called the differential code. It constitutes an $\left[n,\ell \left(G-D\right)-\ell \left(G\right)+\deg D,d^{⊥}\right]$ code, where $d^{⊥}\leq \deg\left(G\right)-\left(2g-2\right)$, with deg(G)−(2g−2) being its designed distance.

Ref. [20] (Proposition 8.1.2) provides an explicit description of the differential code as a functional code
$$C_{\Omega }\left(D,G\right)=C_{L}\left(D-G+\left(dt\right)-\left(t\right)\right),$$
where *t* is an element of $F_{q^{2}}\left(H_{q}\right)$ such that $v_{Q_{i}}\left(t\right)=1$ for all $i\in \{1,\ldots ,q^{3},\infty \}$. If G=sP and $D=Q_{1}+\cdots +Q_{q^{3}}$, then t=R(x,y) is a good choice, with
$$\left(dt\right)=\left(-dx\right)=\left(2g-2\right)Q_{\infty }=\left(q-2\right)\left(q+1\right)Q_{\infty },$$
see [20] (Lemma 6.4.4). Then, (Equation 8) implies the following proposition:

**Proposition 8.** 

$$C_{\Omega }\left(D,sP\right)=C_{L}\left(D,\left(q^{3}+q^{2}-q-2\right)Q_{\infty }-sP\right).□$$



The computation of $C_{\Omega }\left(\tilde{D},sP\right)$ is more complicated. We claim the next results for the prime powers q≡2(mod3), since the proofs are rather transparent in this case. We are certain that they hold for q≡1(mod3) as well. Our opinion is supported by numerical experiments with q≤8.

**Lemma 5.** 
*Assume q≡2(mod3) and define the $F_{q^{2}}$-rational function*

$$T=\frac{1}{3}\left(\frac{\ell_{1}^{q^{2}}}{\ell_{2}}+\frac{\ell_{2}^{q^{2}}}{\ell_{3}}+\frac{\ell_{3}^{q^{2}}}{\ell_{1}}\right).$$


*Then,*

$$d\left(\frac{R}{{\left(\ell_{1}\ell_{2}\ell_{3}\right)}^{\frac{q^{2}-q+1}{3}}}\right)=-\left(\frac{T}{{\left(\ell_{1}\ell_{2}\ell_{3}\right)}^{\frac{q^{2}-q+1}{3}}}\right)dx.$$



**Proof.** We have $d\ell_{i}={\left(a_{i}-x\right)}^{q}dx$, and
$$\begin{array}{cc} \ell_{i}^{q^{2}}-\ell_{i+1} & =a_{i}^{q^{3}}x^{q^{2}}-y^{q^{2}}-b_{i}^{q^{3}}-\left(a_{i+1}^{q}x-y-b_{i+1}^{q}\right)\\ \; & =a_{i+1}^{q}\left(x^{q^{2}}-x\right)-\left(y^{q^{2}}-y\right)\\ \; & =a_{i+1}^{q}R\left(x,y\right)-x^{q}R\left(x,y\right)\\ \; & ={\left(a_{i+1}-x\right)}^{q}R\left(x,y\right). \end{array}$$In one line,
(9)$$\begin{array}{c} \frac{\left(a_{i+1}-x\right){)}^{q}}{\ell_{i+1}}=\frac{\ell_{1}^{q^{2}}/\ell_{2}-1}{R(x,y)}. \end{array}$$Hence,
$$\begin{array}{cc} d(\ell_{1}\ell_{2}\ell_{3}) & =\ell_{1}\ell_{2}\ell_{3}\cdot \left(\frac{{\left(a_{1}-x\right)}^{q}}{\ell_{1}}+\frac{{\left(a_{2}-x\right)}^{q}}{\ell_{2}}+\frac{{\left(a_{3}-x\right)}^{q}}{\ell_{3}}\right)\;dx\\ \; & =\ell_{1}\ell_{2}\ell_{3}\cdot \left(\frac{\ell_{1}^{q^{2}}/\ell_{2}-1}{R}+\frac{\ell_{2}^{q^{2}}/\ell_{3}-1}{R}+\frac{\ell_{3}^{q^{2}}/\ell_{1}-1}{R}\right)\;dx\\ \; & =\frac{\ell_{1}\ell_{2}\ell_{3}}{R}\left(3T-3\right)dx. \end{array}$$This implies
$$\begin{array}{c} d\left(R{\left(\ell_{1}\ell_{2}\ell_{3}\right)}^{\frac{-q^{2}+q-1}{3}}\right)=\left(-{\left(\ell_{1}\ell_{2}\ell_{3}\right)}^{\frac{-q^{2}+q-1}{3}}\right)dx+\\ R\left(-\frac{1}{3}{\left(\ell_{1}\ell_{2}\ell_{3}\right)}^{\frac{-q^{2}+q-4}{3}}\right)\frac{\ell_{1}\ell_{2}\ell_{3}}{R}\left(3T-3\right)dx. \end{array}$$By easy cancellation
$$\begin{array}{cc} d\left(R{\left(\ell_{1}\ell_{2}\ell_{3}\right)}^{\frac{-q^{2}+q-1}{3}}\right) & =\left(-{\left(\ell_{1}\ell_{2}\ell_{3}\right)}^{\frac{-q^{2}+q-1}{3}}\right)dx+-\frac{1}{3}{\left(\ell_{1}\ell_{2}\ell_{3}\right)}^{\frac{-q^{2}+q-1}{3}}\left(3T-3\right)dx\\ \; & =-\left(\frac{T}{{\left(\ell_{1}\ell_{2}\ell_{3}\right)}^{\frac{q^{2}-q+1}{3}}}\right)dx. \end{array}$$□

**Lemma 6.** 
*Assume q≡2(mod3) and define the $F_{q^{2}}$-rational functions*

$$T=\frac{1}{3}\left(\frac{\ell_{1}^{q^{2}}}{\ell_{2}}+\frac{\ell_{2}^{q^{2}}}{\ell_{3}}+\frac{\ell_{3}^{q^{2}}}{\ell_{1}}\right)\;and\;R_{1}=\frac{R}{{\left(\ell_{1}\ell_{2}\ell_{3}\right)}^{\frac{q^{2}-q+1}{3}}}.$$


*Let G be a divisor of $F_{q^{2}}\left(H_{q}\right)$ whose support is disjoint from the support of $\tilde{D}$. Then,*

$$L\left(\tilde{D}-G+\left(dR_{1}\right)-\left(R_{1}\right)\right)=L\left(\frac{\left(q^{2}-1\right)\left(q+1\right)}{3}P-G\right)\cdot \frac{{\left(\ell_{1}\ell_{2}\ell_{3}\right)}^{\frac{q^{2}-1}{3}}}{T}.$$



**Proof.** We have
$$\begin{array}{cc} \tilde{D}-G+\left(dR_{1}\right)-\left(R_{1}\right) & =\tilde{D}-G+\left(T\right)-\frac{q^{2}-q+1}{3}\left(\ell_{1}\ell_{2}\ell_{3}\right)+\left(dx\right)\\ \; & \;-\left(R\right)+\frac{q^{2}-q+1}{3}\left(\ell_{1}\ell_{2}\ell_{3}\right)\\ \; & =\tilde{D}-G+\left(T\right)+\left(dx\right)-\left(R\right)\\ \; & =Q_{\infty }+q^{3}Q_{\infty }+\left(2g-2\right)Q_{\infty }-G+\left(T\right)\\ \; & =\left(q^{2}-1\right)\left(q+1\right)Q_{\infty }-G+\left(T\right)\\ \; & =\frac{\left(q^{2}-1\right)\left(q+1\right)}{3}P-\left({\left(\ell_{1}\ell_{2}\ell_{3}\right)}^{\frac{q^{2}-1}{3}}\right)-G+\left(T\right). \end{array}$$For Riemann–Roch spaces, the results follow. □

**Lemma 7.** 
*For any i,j∈{1,2,3}, we have*

$$\left(\frac{\ell_{i}}{\ell_{j}}\right)\left(Q_{\infty }\right)=1.$$



**Proof.** We use the local expansion $\tau \left(t\right)=\left(t:1:t^{q+1}+\cdots \right)$ of $H_{q}$ at $Q_{\infty }$. The dots represent terms of a higher degree.
$$\left(\frac{\ell_{i}}{\ell_{j}}\right)\left(\tau \left(t\right)\right)=\frac{a_{i}^{q}t-1-b_{i}^{q}\left(t^{q+1}+\cdots \right)}{a_{j}^{q}t-1-b_{j}^{q}\left(t^{q+1}+\cdots \right)},$$
which implies
$$\left(\frac{\ell_{i}}{\ell_{j}}\right)\left(Q_{\infty }\right)=\left(\frac{\ell_{i}}{\ell_{j}}\right)\left(\tau \left(0\right)\right)=1.$$□

**Lemma 8.** 
*Assume q≢0(mod3) and define the $F_{q^{2}}$-rational functions*

T=13ℓ1q2ℓ2+ℓ2q2ℓ3+ℓ3q2ℓ1andT1=(ℓ1ℓ2ℓ3)q2−13T.


*Then, $T_{1}\left(Q_{\infty }\right)=1$.*


**Proof.** Since
$$\frac{\ell_{i}^{q^{2}}}{\ell_{i+1}{\left(\ell_{1}\ell_{2}\ell_{3}\right)}^{\frac{q^{2}-1}{3}}}$$
is the product of terms such as $\ell_{i}/\ell_{j}$, it takes the value of 1 at $Q_{\infty }$. This implies $\left(1/T_{1}\right)\left(Q_{\infty }\right)=1$. □

Before stating our main result on differential codes, we remind the reader that two linear codes $C_{1},C_{2}$ are monomially equivalent if $C_{2}=\mu ★C_{1}$ for some multiplier vector μ.

**Theorem 4.** 
*Assume q≡2(mod3) and define the $F_{q^{2}}$-rational functions*

T=13ℓ1q2ℓ2+ℓ2q2ℓ3+ℓ3q2ℓ1andT1=(ℓ1ℓ2ℓ3)q2−13T.


*Let G be a divisor of $F_{q^{2}}\left(H_{q}\right)$, whose support is disjoint from the support of $\tilde{D}$. Define $\mu_{i}=T_{1}\left(Q_{i}\right)$ for $i\in \{1,\ldots ,q^{3},\infty \}$ and write $\mu =(\mu_{i})$. Then, all entries $\mu_{i}\in F_{q^{2}}^{*}$, and*

$$C_{\Omega }\left(\tilde{D},G\right)=\mu ★C_{L}\left(\tilde{D},\frac{\left(q^{2}-1\right)\left(q+1\right)}{3}P-G\right).$$



**Proof.** If $i\in \{1,\ldots ,q^{3}\}$, then $\ell_{i}^{q^{2}}\left(Q_{i}\right)=\ell_{i+1}\left(Q_{i}\right)$. Therefore, $T(Q_{i})=1$ and $T_{1}\left(Q_{i}\right)$ is a well-defined non-zero element in $F_{q}$. Lemma 8 implies $T_{1}\left(Q_{\infty }\right)=1$. The theorem follows from Lemma 6. □

**Corollary 1.** 

$$C_{\Omega }\left(\tilde{D},sP\right)=\mu ★C_{L}\left(\tilde{D},\left(\frac{\left(q^{2}-1\right)\left(q+1\right)}{3}-s\right)P\right).$$



## 6. Hermitian Subfield Subcodes from Degree-Three Places

In this section, we study the subfield subcodes of $C_{L}\left(D,sP\right)$. As before, *q* is a prime power, s≥0 integer, and *P* is a place of degree three of the Hermitian curve $H_{q}$. The divisor $D=Q_{1}+\cdots +Q_{n}$, $n=q^{3}$, is defined as the sum of the $F_{q^{2}}$-rational affine places of $H_{q}$. The rational place at infinity is $Q_{\infty }$ and $\tilde{D}=D+Q_{\infty }$.

### 6.1. Trace Maps of Hermitian Functions and Hermitian Codes

We collect properties of the maps $z\mapsto z^{q}+z$ and $z\mapsto z^{q}-z$, where *z* is either a field element, a function, or a vector. We refer to $z^{q}+z$ as the trace of *z*, and to the map itself as the trace map $\mathrm{Tr}={\mathrm{Tr}}_{F_{q^{2}}/F_{q}}$. Clearly, Tr is linear over $F_{q}$.

**Lemma 9.** 
*Consider a positive divisor $G_{1}$. The trace map satisfies the following properties:*
*(i)* 
*For any function $f\in L(G_{1})$, its trace lies within $L(qG_{1})$, implying $\mathrm{Tr}\left(L\left(G_{1}\right)\right)\subseteq L\left(qG_{1}\right)$.*
*(ii)* 
*Similarly, for any codeword $c\in C_{L}\left(D,G_{1}\right)$, its trace resides in $C_{L}\left(D,qG_{1}\right)$.*
*(iii)* 
*$\mathrm{Tr}(C_{L}\left(D,G_{1}\right))$ is an $F_{q}$-linear subspace of $C_{L}\left(D,qG_{1}\right)\cap F_{q}^{n}$.*



**Proof.** Since $G_{1}\geq 0$, we have $L\left(G_{1}\right),L{\left(G_{1}\right)}^{q}\leq L\left(qG_{1}\right)$; hence, (i) holds. Then, (i) implies (ii), and (iii) follows trivially. □

**Proposition 9.** 
*Let $G_{1}$ be a positive divisor that satisfies $\deg G_{1}<n/q$. Then, $\mathrm{Tr}(C_{L}\left(D,G_{1}\right))$ is an $F_{q}$-linear subfield subcode of $C_{L}\left(D,qG_{1}\right)$. Its dimension is*

$$\dim_{F_{q}}\left(\mathrm{Tr}\left(C_{L}\left(D,G_{1}\right)\right)\right)=2\dim_{F_{q^{2}}}\left(L\left(G_{1}\right)\right)-1.$$



**Proof.** $\mathrm{Tr}(C_{L}\left(D,G_{1}\right))$ is an $F_{q}$-linear subfield subcode by Lemma 9. The trace map Tr and the evaluation map ${\mathrm{ev}}_{D}$ commute, and by $\deg(G_{1})<n$, ${\mathrm{ev}}_{D}$ is injective. Define the $F_{q}$-linear map
$$\tau :L\left(G_{1}\right)\to C_{L}\left(D,qG_{1}\right)\cap F_{q}^{n},\;f\mapsto {\mathrm{ev}}_{D}\left(\mathrm{Tr}\left(f\right)\right).$$On the one hand,
dimFq(L(G1))=2dimFq2(L(G1))=dimIm(τ)+dimker(τ).We have to show that ker(τ)=1. Define $\varepsilon \in F_{q^{2}}$ such that ε=1 if *q* is even and $\varepsilon =g^{(q+1)/2}$ if *q* is odd and *g* is a primitive element in $F_{q^{2}}$. Then, $\varepsilon^{q-1}=-1$. For the rational function $f\in F_{q^{2}}\left(H\left(q\right)\right)$, we have
$$\begin{array}{cc} f\in \ker(\tau ) & \Rightarrow f^{q}+f=0\\ \; & \Rightarrow {\left(\varepsilon f\right)}^{q}=\varepsilon \;f\\ \; & \Rightarrow \varepsilon f\in F_{q}\\ \; & \Rightarrow f\in \varepsilon^{-1}F_{q}. \end{array}$$This finishes the proof. □

### 6.2. An Explicit Subfield Subcode

In this subsection, we study a subfield subcode of $C_{L}\left(D,\left(q^{2}-q+1\right)P\right)$. As $q^{2}-q+1=\left(q-1\right)\left(q+1\right)-\left(q-1\right)$, one has
$$U_{q^{2}-q+1,i}=\frac{\ell_{i}^{q}\ell_{i+2}}{\ell_{i+1}\ell_{i+2}^{q}}.$$
The vector space $W_{q^{2}-q+1}\leq L\left(\left(q^{2}-q+1\right)P\right)$ consists of the functions
$$W_{q^{2}-q+1,\alpha }=\alpha \frac{\ell_{1}^{q}\ell_{3}}{\ell_{2}\ell_{3}^{q}}+\alpha^{q^{2}}\frac{\ell_{2}^{q}\ell_{1}}{\ell_{3}\ell_{1}^{q}}+\alpha^{q^{4}}\frac{\ell_{3}^{q}\ell_{2}}{\ell_{1}\ell_{2}^{q}},\;\alpha \in F_{q^{6}}.$$

For rational functions $f,g\in F_{q^{6}}\left(H_{q}\right)$, we introduce the relation
$$f\approx g⟺f\left(Q_{i}\right)=g\left(Q_{i}\right)\;\mathrm{for}\;\mathrm{all}\;i\in \{1,\ldots ,q^{3},\infty \}.$$

This is clearly an equivalence relation, which can be also written in terms of the principal divisor
$$f\approx g⟺\left(f-g\right)\geq \tilde{D},$$
or in terms of the evaluation map
$$f\approx g⟺{\mathrm{ev}}_{\tilde{D}}\left(f\right)={\mathrm{ev}}_{\tilde{D}}\left(g\right).$$

**Lemma 10.** 
*(i)* 
*${\left(U_{q^{2}-q+1,i}\right)}^{q}\approx U_{q^{2}-q+1,i+2}$.*
*(ii)* 
*${\left(W_{q^{2}-q+1,\alpha }\right)}^{q}\approx W_{q^{2}-q+1,\alpha^{q^{3}}}$.*



**Proof.** Lemma 7 implies $U_{q^{2}-q+1,i}\left(Q_{\infty }\right)=1$. In the proof of Lemma 5, we have seen that $\ell_{i}^{q^{2}}-\ell_{i+1}={\left(a_{i+1}-x\right)}^{q}R\left(x,y\right)$. Therefore, $\left(\ell_{i}^{q^{2}}-\ell_{i+1}\right)\left(Q_{i}\right)=0$ for all $i\in \{1,\ldots ,q^{3}\}$. This shows
$$\begin{array}{cc} {\left(U_{q^{2}-q+1,i}\right)}^{q}\left(Q_{i}\right) & =\left(\frac{\ell_{i}^{q^{2}}\ell_{i+2}^{q}}{\ell_{i+1}^{q}\ell_{i+2}^{q^{2}}}\right)\left(Q_{i}\right)\\ \; & =\left(\frac{\ell_{i+1}\ell_{i+2}^{q}}{\ell_{i+1}^{q}\ell_{i}}\right)\left(Q_{i}\right)\\ \; & =U_{q^{2}-q+1,i+2}\left(Q_{i}\right) \end{array}$$This proves (i). For (ii):
$$\begin{array}{cc} {\left(W_{q^{2}-q+1,\alpha }\right)}^{q} & ={\left(\alpha U_{q^{2}-q+1,1}+\alpha^{q^{2}}U_{q^{2}-q+1,2}+\alpha^{q^{4}}U_{q^{2}-q+1,3}\right)}^{q}\\ \; & \approx \alpha^{q}U_{q^{2}-q+1,3}+\alpha^{q^{3}}U_{q^{2}-q+1,1}+\alpha^{q^{5}}U_{q^{2}-q+1,2}\\ \; & =\alpha^{q^{3}}U_{q^{2}-q+1,1}+{\left(\alpha^{q^{3}}\right)}^{q^{2}}U_{q^{2}-q+1,2}+{\left(\alpha^{q^{3}}\right)}^{q^{4}}U_{q^{2}-q+1,3}\\ \; & =W_{q^{2}-q+1,\alpha^{q^{3}}}. \end{array}$$□

**Proposition 10.** 
*The set*

$$\tilde{W}=\left\{{\mathrm{ev}}_{D}\left(W_{q^{2}-q+1,\alpha }\right)|\alpha \in F_{q^{3}}\right\}$$

*is a three-dimensional $F_{q}$-linear subfield subcode of $C_{L}\left(D,\left(q^{2}-q+1\right)P\right)$.*


**Proof.** Lemma 10(ii) implies that ${\mathrm{ev}}_{D}\left(W_{q^{2}-q+1,\alpha }\right)$ has $F_{q}$-entries if and only if $\alpha^{q^{3}}=\alpha$. □

### 6.3. Main Result and a Conjecture

**Theorem 5.** 
*Let q≥3 be a prime power, $n=q^{3}$, $D=Q_{1}+\cdots +Q_{n}$ be the sum of rational affine places of $F_{q^{2}}\left(H_{q}\right)$, and P be a place of degree three. The dimension of the subfield subcode of the one-point Hermitian code is*

$$\dim C_{L}{\left(D,sP\right)|}_{F_{q}}\geq \left\{\begin{array}{cc} 7 & \mathrm{for}\;s=2g=q(q-1),\\ 10 & \mathrm{for}\;s=2g+1=q^{2}-q+1. \end{array}\right.$$



**Proof.** Set $G_{1}=\left(q-1\right)P$. By Proposition 9,
$$T={\mathrm{ev}}_{D}\left(\mathrm{Tr}\left(L\left(G_{1}\right)\right)\right)$$
is an $F_{q}$-linear subspace in $C_{L}{\left(D,q\left(q-1\right)P\right)|}_{F_{q}}$. Since dim(L((q−1)P))=4, T has dimension seven. This proves $\dim C_{L}{\left(D,q\left(q-1\right)P\right)|}_{F_{q}}\geq 7$.Let $\tilde{W}$ be the three-dimensional $F_{q}$-linear subfield subcode of $C_{L}\left(D,\left(q^{2}-q+1\right)P\right)$ given in Proposition 10. We show that $T\cap \tilde{W}=\left\{0\right\}$; the inequality $\dim C_{L}\left(D,\left(q^{2}-q+1\right)P\right){|}_{F_{q}}\geq 10$ will follow. On the one hand,
$$\tilde{W}\leq {\mathrm{ev}}_{D}\left(W_{q^{2}-q+1}\right).$$On the other hand, using Theorem 3, we have
$$T\leq {\mathrm{ev}}_{D}\left(L\left(q\left(q-1\right)P\right)\right)={\mathrm{ev}}_{D}\left(\underset{t\in H(P),\;t\leq q(q-1)}{⨁}W_{t}\right).$$As ${\mathrm{ev}}_{D}$ is injective on $L(\left(q^{2}-q+1\right)P)$, and
$$\left(\underset{t\in H(P),\;t\leq q(q-1)}{⨁}W_{t}\right)\cap W_{q^{2}-q+1}=\left\{0\right\},$$
we obtain $T\cap \tilde{W}=\left\{0\right\}$. This completes the proof. □

Our proof was constructive, we used the subfield subcodes given explicitly in the previous subsections. Based on computer calculations for small *q*, we have the following conjecture.

**Conjecture 1.** 
*If q≥4, then equalities hold in Theorem 5.*


The claim of the conjecture has some equivalent formulations.

**Proposition 11.** 
*The following are equivalent.*
*(i)* 
*$\dim C_{L}\left(D,\left(q^{2}-q\right)P\right){|}_{F_{q}}=7$.*
*(ii)* 
*$\dim C_{L}\left(D,\left(q^{2}-q-1\right)P\right){|}_{F_{q}}=1$.*
*(iii)* 
*$\dim C_{L}{\left(D,sP\right)|}_{F_{q}}=1$ for all $0\leq s\leq 2g-1=q^{2}-q-1$.*



**Proof.** We use the notation of the proof of Theorem 5. Assume (i). We have $L\left(\left(q-1\right)P\right)=W_{0}⊕W_{q-1}$. Moreover, T is an $F_{q}B$-module that decomposes into the direct sum of a one-dimensional submodule and a six-dimensional submodule. Note that any non-trivial irreducible $F_{q}B$-module has dimension six. Since $T\cap C_{L}\left(D,\left(q^{2}-q-1\right)P\right)$ is a proper submodule, the only possibility is that it is one-dimensional over $F_{q}$. (ii) follows. Trivially, (ii) implies (iii). Let us now assume (iii).
$$\dim_{F_{q}}C_{L}\left(D,\left(q^{2}-q\right)P\right)/C_{L}\left(D,\left(q^{2}-q-1\right)P\right)=6,$$
and therefore,
$$\dim_{F_{q}}C_{L}\left(D,\left(q^{2}-q\right)P\right){{|}_{F_{q}}/C_{L}\left(D,\left(q^{2}-q-1\right)P\right)|}_{F_{q}}\leq 6.$$This implies $\dim C_{L}\left(D,\left(q^{2}-q\right)P\right){|}_{F_{q}}\leq 7$. Together with Theorem 5, we have (i). □

We have a partial result related to case (iii) of Proposition 11.

**Proposition 12.** 
*$\dim C_{L}{\left(D,sP\right)|}_{F_{q}}=1$ for all $0\leq s\leq \frac{2}{3}g$.*


**Proof.** Fix an arbitrary integer *s* in the range $0\leq s<\frac{2}{3}g$ and consider a generic element $\left(c_{1},\ldots ,c_{q^{3}}\right)\in C_{q}\left(s\right)$. This corresponds to a function *g* in L(sP) such that $c_{i}=g\left(Q_{i}\right)$ is an element of $F_{q}$ for each $i=1,\ldots ,q^{3}$. We note that there exists a $\gamma \in F_{q}$ such that at least $q^{2}$ of the $c_{i}$ values is equal to γ. In other words, the function g−γ is in L(sP) and has at least $q^{2}$ zeros on $H_{q}$. However, a non-zero function in L(sP) cannot have more than deg(G)≤2g=q(q−1) zeros, leading us to conclude that g−γ must be the zero function. This implies that every $c_{i}$ is equal to γ, and hence $C_{L}{\left(D,sP\right)|}_{F_{q}}$ consists of constant vectors. This completes the proof. □

## 7. Conclusions

In summary, our research has uncovered important properties of the family of Hermitian subfield subcodes associated with degree-three places. We achieved this by precisely determining the dimension of these codes for certain parameters and providing explicit bases for the corresponding Riemann–Roch spaces. Moreover, we conducted experiments aimed at calculating the exact dimension of the underlying family of codes across a broad spectrum of parameters. This process has contributed to the reformulation of certain conjectures, with some being proven. Additionally, we have established lower bounds on the dimension of Hermitian subfield subcodes associated with the divisor sP, where *P* is a degree-three Hermitian place, for specific cases such as $0\leq s\leq \frac{2}{3}g$, s=2g, and s=2g+1, utilizing the bases of the underlying family of codes. Our motivation to explore the properties of Hermitian subfield subcodes stems from their potential as a family of AG codes for post-quantum cryptography use. In our future work, we anticipate using the parameters of subfield subcodes of degree-three Hermitian codes to enhance and secure the McEliece cryptosystem.

## Data Availability

No relevant new data were created in this research.

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
