# Peer review of "On the Dimensions of Hermitian Subfield Subcodes from Higher-Degree Places"

_entropy, 2024, doi:10.3390/e26050386_

Round 1
Reviewer 1 Report
Comments and Suggestions for Authors
The presented manuscript concentrates on the research on the family of subfield subcodes of Hermitian codes associated with degree 3. The authors proved the lower bound on the dimension of these codes. The results of this research may affect some important cryptographic applications and have a positive theoretical impact.
In general, the manuscript is written well, with a concentration on pure theoretical results. The main contribution is the series of theorems and propositions supported by proofs.
However, the structure of the manuscript and presentation logic may be improved as follows:
1. I would recommend to pay more attention to describing the ideas rather than mathematical proofs. The manuscript is complex and hard to follow. Perhaps some less important proofs may be moved to the Appendix sections while more detailed comments are added to the main part with the intention to clarify the logic of the presentation.
2. This manuscript does not have a Conclusion. I would recommend that the authors add the proper conclusion.
3. At the end of the Introduction, the authors mentioned computational results. However, they did not provide any in the manuscript. Therefore, the manuscript does not feel complete. Please clarify this moment. In general, simplified examples of the analysed codes will be helpful for readers.
4. Authors mentioned the application of the subfield subcodes of Hermitian codes associated with degree 3 to the McEliece cryptosystem. It would be helpful if the authors could provide some basic thoughts on this subject and mention how exactly their result will affect this cryptosystem (which has important potential implications in post-quantum cryptography).
5. I would recommend extending the literature review on the related works, which would help to compare the presented work to another existing result.
In conclusion, this work extends our understanding of Hermitian codes, which is valuable from a theoretical point of view. It is almost impossible to carefully verify all the theoretical results in a short time. However, I could not find any flaws in the provided proofs at first glance. I recommend to accept the manuscript after major revision if the authors will successfully improve the presentation of the manuscript and correct the structure.
Comments on the Quality of English LanguageThe quality of English language is acceptable
Author Response
We would like to express our gratitude for the time and effort the Reviewer has put into reviewing our manuscript. We appreciate the insightful comments and suggestions. Below, we provide our responses to each of his/her comments, along with the modifications we have made to the manuscript.
1. Since Reviewer 2 recommended the acceptance of the paper in the present form, we decided to keep the structure of the paper and extend the Introduction, and add a Conclusion.
2. We have added a Conclusion section that summarizes the main contributions, implications, and potential directions for future research.
3. In the Introduction, we included a paragraph on the computational results. Specifically, we explained that the computational evidence supporting the Conjecture does not rely on the theoretical results of this paper. The "HERmitian" package of GAP is well documented, and the computation can be easily verified using the commands "Hermitian_FunctionalCode" and "RestrictVectorSpace".
4. We have expanded the paragraph referring to McEliece cryptosystems in the Introduction, emphasizing that our investigation
will allow for a comparison of the parameters of Hermitian codes with those of other existing codes, such as Goppa codes,
potentially leading to an improvement in the key size of the McEliece cryptosystem.
5. We would like to emphasize that our paper primarily focuses on the theoretical findings concerning the structure of Hermitian Riemann-Roch spaces and their subfield subcodes. We are not aware of further relevant literature on this subject.
Once again, we would like to express our gratitude for your constructive feedback. We believe that the revisions we have made will significantly improve the quality and accessibility of our manuscript.
Reviewer 2 Report
Comments and Suggestions for Authors
The construction of Hermitian codes based on Hermitian curves over finite fields is thorughly studied and the authors examine the structure of subfield subcodes and trace codes. The authors state a plausible conjecture for the dimension of some subfield subcodes of Hermitian codes. This is by itself an open via for research on Hermitian codes. The paper is quite sound and self contained, thus I strongly recommend its acceptation in its current form.
Author Response
We would like to express our gratitude for the time and effort the Reviewer has put into reviewing our manuscript. We appreciate his/her valuable remarks. Reviewer 1 asked for a Conclusion section (included at the end of the paper) and more detail about the computational results (included in the Introduction).